# A Welding Defect Detection Model Based on Hybrid-Enhanced Multi-Granularity Spatiotemporal Representation Learning

**DOI:** 10.3390/s25154656

**Published:** 2025-07-27

**Authors:** Chenbo Shi, Shaojia Yan, Lei Wang, Changsheng Zhu, Yue Yu, Xiangteng Zang, Aiping Liu, Chun Zhang, Xiaobing Feng

**Affiliations:** 1College of lntelligent Equipment, Shandong University of Science and Technology, Taian 271019, China; skd996523@sdust.edu.cn (C.S.); 202383230010@sdust.edu.cn (S.Y.); 202283230040@sdust.edu.cn (L.W.); zcs@sdust.edu.cn (C.Z.); 202383230013@sdust.edu.cn (Y.Y.); zangxt@sdust.edu.cn (X.Z.); 2Beijing Botsing Technology Co., Ltd., Beijing 100176, China; lap@botsing.net

**Keywords:** image interference, porosity defect, deep learning, key interference-free frames, multi-granularity spatiotemporal features

## Abstract

Real-time quality monitoring using molten pool images is a critical focus in researching high-quality, intelligent automated welding. To address interference problems in molten pool images under complex welding scenarios (e.g., reflected laser spots from spatter misclassified as porosity defects) and the limited interpretability of deep learning models, this paper proposes a multi-granularity spatiotemporal representation learning algorithm based on the hybrid enhancement of handcrafted and deep learning features. A MobileNetV2 backbone network integrated with a Temporal Shift Module (TSM) is designed to progressively capture the short-term dynamic features of the molten pool and integrate temporal information across both low-level and high-level features. A multi-granularity attention-based feature aggregation module is developed to select key interference-free frames using cross-frame attention, generate multi-granularity features via grouped pooling, and apply the Convolutional Block Attention Module (CBAM) at each granularity level. Finally, these multi-granularity spatiotemporal features are adaptively fused. Meanwhile, an independent branch utilizes the Histogram of Oriented Gradient (HOG) and Scale-Invariant Feature Transform (SIFT) features to extract long-term spatial structural information from historical edge images, enhancing the model’s interpretability. The proposed method achieves an accuracy of 99.187% on a self-constructed dataset. Additionally, it attains a real-time inference speed of 20.983 ms per sample on a hardware platform equipped with an Intel i9-12900H CPU and an RTX 3060 GPU, thus effectively balancing accuracy, speed, and interpretability.

## 1. Introduction

As a core industrial technology, welding is extensively applied in advanced manufacturing sectors such as aerospace, shipbuilding, rail transit, and machinery, playing a vital role in national economic development and technological advancement [1]. Achieving stable and high-quality welds—characterized by defect-free surfaces, consistent geometry, and reliable mechanical properties—requires optimal process design, precise parameter control, skilled operation, and robust quality assurance. However, due to the complex, dynamic, and nonlinear nature of the welding process, various defects are often unavoidable, and detecting them in real time remains challenging [2]. Experienced welders manually detect defects by observing the molten pool. However, prolonged observation causes fatigue, slower responses, and lower quality, while welding fumes also pose health risks. These challenges have driven global research efforts toward automated molten pool vision systems that can continuously monitor welding quality and detect defects in real time, thereby improving efficiency and safety.

The current research field of welding defect recognition and monitoring in the welding process is divided into two main categories: the first method focuses on welding defect detection using a single-frame molten pool image; the second method realizes welding defect recognition by analyzing the spatiotemporal dynamic information embedded in the molten pool image sequence.

Welding defect detection based on single-frame molten pool images typically relies on analyzing static frames captured at a single time point. Traditional machine learning methods employ algorithms such as decision trees, support vector machines (SVMs), and random forests (RFs) to simulate expert decision-making [3,4,5,6,7]. These methods are interpretable but depend on expert knowledge and precise camera–robot alignment. End-to-end CNNs eliminate manual feature design and have achieved improved accuracy and efficiency through architectural enhancements [2,8,9,10,11,12,13], while hybrid approaches that fuse deep and handcrafted features further bolster interpretability and robustness [14,15]. However, visual similarities across welding conditions continue to limit the reliability of single-frame analysis in real-world scenarios. Compared to single-frame methods, multi-frame methods utilize time-series molten pool images, providing richer dynamic information for robust weld quality monitoring. Some scholars have utilized typical timing processing methods to establish welding defect detection models; for example, Lu et al. [1] used an LSTM-based prediction–classification framework to model temporal morphology changes up to ten frames ahead, while Chen et al. [16] fused CNN–LSTM visual features with arc signal time–frequency features for improved GTAW state recognition. Liu et al. [17] proposed a lightweight 3DCNN (3DSMDA-Net) that decomposes 3D convolutions and incorporates multidimensional attention to reduce computation without sacrificing accuracy. Despite these advances, real-time deployment remains challenging, prompting frame-fusion strategies such as that by Jiao et al. [18], who combined adjacent frames into a single image for efficient 2D-CNN processing.

In addition, in the actual welding process, welding defect detection using molten pool images still faces many challenges. As shown in Figure 1a,b, the interclass similarity problem poses a major difficulty in recognizing welding defects. Due to the extremely complex heat transfer and molten metal flow processes in the weld pool, the normal pool and the defective (e.g., porosity) pool are highly similar regarding surface spot distribution, texture characteristics, and overall appearance. This similarity makes it very easy for the traditional single-frame image analysis method to misjudge the welding state, thereby reducing the accuracy and reliability of the identification. Meanwhile, as shown in Figure 1c,d, the problem of intraclass variability also hinders the stability of molten pool image analysis. Even under the same welding condition (e.g., a normal molten pool), the molten pool images may be affected by the interference of spatter particles, changes in the intensity of the molten pool flow, and small differences in the shooting angle and illumination conditions, resulting in large differences in the morphology, texture, and brightness distribution of the molten pool. This significant intraclass variability increases the difficulty of the model in accurately identifying the steady state, which in turn reduces the stability of the actual detection. To address these issues, Hong et al. [8] address these challenges with LDI-MSARL-net, which employs multi-granularity spatiotemporal attention, a dual-branch global–local fusion, and dynamic enhancement to improve feature discrimination. Despite such progress, efficiently and reliably overcoming both interclass similarity and intraclass variability in real-world molten pool images remains an open challenge.

Previous studies have shown the potential value of applying spatiotemporal feature information to weld defect monitoring. However, some existing detection methods still have some problems: (1) The “black box” nature of deep learning makes it difficult to trace the decision logic of the model, while welding defect detection needs to meet the strong interpretability requirements of industrial scenarios (e.g., defect attribution analysis). Techniques like CAM or Grad-CAM [19] highlight high-level activations but cannot link them to physical molten pool characteristics (e.g., temperature or geometry). Fusing handcrafted descriptors—such as LBP for surface roughness [20] and HOG for contour aberrations—with CNN features (e.g., ResNet-50 activation maps [21]) via multimodal alignment preserves interpretability while reducing overfitting. (2) The problem of molten pool image interference (e.g., the reflected spot of the laser on the spatter is misclassified as a stomatal defect) in a complex welding scene is prominent. Some attentional mechanisms can overcome the interclass similarity problem of defects on a spatial scale, but in contrast, learning key unobstructed frames in time-series images may bring better results for recognizing such images. Considering both the interclass similarity of defects as well as the intraclass diversity problem, we use a multi-granular spatiotemporal attention mechanism for molten pool time-series images.

Inspired by previous research, we propose a multi-granularity spatiotemporal attention learning network (HIE-MSTARLNet). This network integrates temporal deep learning features with long-term dynamic information to address these challenges. The main contributions of this paper are summarized as follows.

We propose an innovative welding defect detection method that integrates the selection of key interference-free frames into a multi-granularity spatiotemporal attention mechanism, thereby overcoming the limitations of existing approaches in extracting critical information from highly redundant, interference-prone molten pool time-series images;Simultaneously aggregating long-term dynamic information of fusion pool time-series images and multi-level semantic features in the spatial dimension of time-series deep learning features is achieved, effectively solving the problem of insufficient interpretability and the problem of interclass similarity and intraclass diversity of deep learning;Based on the collected data on actual weld porosity defects, the effectiveness of the proposed method is verified through careful evaluation experiments, comparison experiments with typical methods, and visualization analysis.

## 2. Methods

The methodology used in this paper begins by using images captured by a molten pool vision system that are first pre-processed; then, a sequence of 8 consecutive frames is fed into a defect monitoring model. The model recognizes the type of defect and issues a warning if necessary.

### 2.1. Molten Pool Vision System

As shown in Figure 2, this study uses the same melting pool vision system as [22]. The molten pool vision system comprises a molten pool camera, a trackless crawling welding robot, and an industrial control computer. The molten pool camera captures videos at a pixel resolution of 640 × 512 with a frame rate of up to 640 FPS. It is controlled by the industrial control computer, allowing for flexible recording. All experiments were conducted on a GMAW welding system.

### 2.2. HIE-MSTARLNet Architecture

This section presents HIE-MSTARLNet, our novel framework that fuses long-term handcrafted spatial cues with a unified multi-granularity attention mechanism to achieve efficient, interpretable spatiotemporal feature learning. Compared to the defect detection model [22] based on multiscale feature fusion for melting pool videos, HIE-MSTARLNet makes several improvements by adding long-term spatial structural features (handcrafted features) extracted from edge history images, replacing the cascade structure of multiscale feature fusion and hybrid attention mechanisms with a multi-granularity attentional feature aggregation model, incorporating the selection of key unobstructed frames, and introducing an attention-based fusion layer to enhance the integration of deep and handcrafted features. The input is consistent with the defect detection model based on the multiscale feature fusion of melting pool videos, which processes the video sequence as input. The video sequence can be represented as $A\in R^{NT\times C\times H\times W}$, where N is the batch size, T is the time dimension, C is the number of channels, and H and W are the spatial dimensions.

The structure of HIE-MSTARLNet is shown in Figure 3, which is mainly composed of two branches: the Temporal Shift–Multi-granularity Feature Aggregation (TS-MAFA) branch and the Auxiliary Feature Extractor (AFE) branch. The raw molten pool image sequence is sent to the TS-MAFA branch to extract deep learning features, while handcrafted features are extracted in the AFE branch. After feature extraction, the features from both branches are concatenated and then passed through an attention fusion layer that adaptively weights and aggregates the complementary information from both modalities, improving feature alignment and overall discriminative capability. The fused feature vector is then fed into a fully connected layer for final classification. In the TS-MAFA branch, the image sequence is first fed into the MobileNetV2 backbone with unidirectional TSM to extract features, followed by the MAFA module to filter key non-obscuring frames and compute attention maps at different granularity levels, thus capturing both the coarse-grained global structure and fine-grained local details [23]. In the AFE branch, the Marginal Edge History Image (MEHI) is first computed, then HOG [24] and SIFT [25] features are extracted from the MEHI, each reduced to 100 dimensions using PCA, and concatenated. These handcrafted features are combined with the deep features via the attention fusion layer before the final classification, which allows the model to better leverage and interpret multi-source information. Overall, the HIE-MSTARLNet architecture is designed for lightweight, efficient time-series modeling and strong interpretability, meeting requirements for accuracy, real-time performance, and explainability.

#### 2.2.1. Temporal Dynamic Characterization

In this subsection, we embed a unidirectional Temporal Shift Module (TSM) into each MobileNetV2 block. This method captures both pixel-level displacement and high-level motion trends without additional computational costs. Following the structure in [22], the unidirectional TSM after each MobileNetV2 module allows the model to effectively mine information about dynamic changes in molten pool image sequences across time. At shallow layers, pixel-level displacements—arising from subtle surface fluctuations or electrode shifts—are acutely captured, while deep layers focus on overall motion trends, such as flow direction and shape evolution. The inherent feature extraction strength of MobileNetV2 [26], combined with temporal channel shifting, enables both shallow and deep features to interact temporally. By shifting a portion of feature channels from previous frames to the current one, the model integrates dynamic information over time, enhancing its ability to perceive changes in motion features. This synergistic temporal encoding supports comprehensive short-term dynamic feature extraction, providing a robust foundation for multi-granularity spatiotemporal representation learning. Importantly, the one-way TSM structure achieves these benefits with negligible computational overhead, making it well-suited for real-time welding defect detection [27].

#### 2.2.2. Long-Term Dynamic Spatial Structural Information

This subsection presents an approach that introduces handcrafted long-term spatial structural features to enhance the interpretability and defect-recognition ability of deep learning models. Specifically, two types of temporal fusion images are processed. First, the Marginal Edge History Image (MEHI) [28] is computed to describe the long-term dynamic content in the molten pool. Edges are detected frame by frame using a dynamic thresholding Canny algorithm, with high thresholds set to μ+1.5σ and low thresholds to 0.5T − high, then superimposed with exponentially decaying weights (λ=0.1) to generate a static edge trajectory map, as shown in Figure 4. This strategy enables the model to incorporate persistent structural cues over time, thereby boosting interpretability and stability in weld defect detection.

Second, the dynamic high-frequency components in the frequency-domain fusion image are computed to describe the long-term dynamic content of the details in the molten pool. Long-term spatial structural features are key components for understanding the spatiotemporal patterns of the dynamic scene, specifically, multi-scale fusion is achieved by Haar wavelet decomposition, in which low-frequency component temporal averaging preserves the background steady state, high-frequency component spatiotemporal maxima projection combines with motion weights to enhance the dynamic details, and an unsharpened mask γ=0.8 is used to sharpen the edges after inverse-transform reconstruction, as shown in Figure 5. To capture the spatial distribution pattern of the target over a long period, after HOG (8 × 8 cellular units, 9-direction gradient histogram) and SIFT (128-dimensional DoG keypoint descriptor) feature extraction, they are, respectively, downscaled to 100 dimensions and spliced into 200-dimensional features by Principal Component Analysis (PCA) and finally constructed with strong characterization ability of the long-term low-dimensional dynamic spatial structure feature vector [29].

#### 2.2.3. Multi-Granularity Attention Feature Aggregation

This subsection introduces the Multi-granularity Attention Feature Aggregation (MAFA) module. MAFA aggregates hierarchical attention across both temporal and spatial scales to accurately capture and discriminate key semantic regions in time-series molten pool images, thereby improving robustness against noise, occlusion, and redundant information. MAFA simultaneously models features at multiple resolutions by computing hierarchical attention weights at both coarse and fine granularities. This enables the network to focus on salient regions across different scales and to suppress frames degraded by blur or occlusion. Inspired by cross attention (CA) [30], we further incorporate a cross-frame attention mechanism that adaptively assigns weights to each temporal frame based on its information content, emphasizing high-quality keyframes while down-weighting low-quality ones (e.g., those affected by occlusion, blur, or noise).

As illustrated in Figure 6, the Multi-granularity Attention Feature Aggregation (MAFA) module processes the input feature tensor $F_{all}\in R^{N\times T\times C\times H\times W}$ in three stages to extract discriminative spatiotemporal semantics at different granularity levels.

First, the input is passed through depth-wise separable convolutions to generate the query (*Q*), key (*K*), and value (*V*) features. These are reshaped and split into $n_{\mathrm{head}}$ attention heads along the channel dimension. Temporal cross-frame attention is then computed to enhance semantic discrimination between frames; it is determined using the following equation:(1)$${\mathrm{Output}}_{\mathrm{head}}=\mathrm{Softmax}\left(\frac{Q_{\mathrm{head}}\cdot K_{\mathrm{head}}^{⊤}}{\sqrt{d_{k}}}\cdot \tau \right)V_{\mathrm{head}}\in R^{N\times T\times n_{\mathrm{head}}\times C/n_{\mathrm{head}}\times \left(H\times W\right)}$$
where $d_{k}=C/n_{\mathrm{head}}$ is a scaling factor and τ is a learnable temperature parameter. The outputs of all attention heads are concatenated along the channel dimension, projected by a 1×1 convolution, and added back to the original input via residual connection. Temporal average pooling is then applied to produce a reference feature $F_{R}\in R^{N\times C\times H\times W}$ for subsequent attention refinement.

Next, CBAM is applied to $F_{R}$ by combining max and average pooling operations across the spatial dimension, followed by convolution and sigmoid activation, to generate spatial attention maps that enhance local feature responses.

Finally, $F_{R}$ and $F_{all}$ are divided into *n* channel groups and pooled at multiple spatial resolutions ($2^{i}$) to capture multi-granularity representations. Each group passes through CBAM to compute attention maps, which are concatenated with the original features and fed into a convolutional fusion layer to generate an attention score vector ($A=(a^{1},a^{2},\ldots ,a^{n})$). These scores are normalized and used to aggregate the corresponding feature groups as follows:(2)$$f=\sum_{i=1}^{n}\mathrm{Softmax}\left(a^{i}\right)⊙x^{i}$$
where ⊙ denotes element-wise multiplication and $x^{i}$ represents the *i*-th granularity group.

Unlike previous methods that apply global multi-scale fusion with hybrid attention, MAFA adopts grouped attention across granularities. Each channel group independently processes features at a specific scale, avoiding redundancy while capturing complementary semantics—ranging from global structures to local details. Temporal dynamics are explicitly encoded via time-averaged reference features, which are absent in prior work that only considers spatial and channel attention. Furthermore, the grouped attention mechanism in MAFA not only reduces computational complexity but also allows each channel group to learn scale-specific attention patterns. Combined with residual connections that promote stable optimization, MAFA effectively captures multi-scale spatiotemporal dependencies in video-based tasks.

### 2.3. Loss Function

The cross-entropy loss is a commonly used loss function in classification problems, especially in deep learning models such as convolutional neural networks (CNNs). It measures the difference between the predicted probability distribution and the actual probability distribution. In a binary classification problem, the output of the model is usually a probability value indicating the likelihood that a sample belongs to a positive class. The cross-entropy loss can be expressed as(3)$$Loss=\frac{1}{N}\sum_{i}^{N}-\left[y_{i}\log\left(p_{i}\right)+\left(1-y_{i}\right)\log\left(1-p_{i}\right)\right]$$
where y denotes the truth value of sample i, where 1 denotes a positive class and 0 denotes a negative class. p denotes the probability that sample i is predicted to be a positive class.

## 3. Experimental Design

### 3.1. Dataset

High-quality datasets are fundamental for algorithm validation. In this study, Gas Metal Arc Welding (GMAW) was employed with two types of welding wire (solid wire and flux-cored wire), and 980 high-strength steel was used as the base material. Welds with varying root gaps and groove angles were prepared, and a molten pool vision system was used to capture videos of both normal and porosity-defect molten pools on 30 welding plates in flat and vertical welding positions [31]. The normal and porosity-defect samples from four of these plates were selected to form the test set. Videos were recorded at a pixel resolution of 640 × 512 and 30 fps, then segmented into 1-s clips with one sample taken every 4 s. Eight frames were evenly extracted from each clip, yielding approximately 600 samples per class—substantially more than classic datasets such as HMDB51 (average of about 101 per category) [32] and UCF101 (average of about 137 per category) [33].

In real welding scenarios, molten pool images include regions of the weld seam, base metal, welding torch, and the molten pool itself; however, the molten pool—the core region for defect detection—occupies only a small portion of each image. High-resolution images contain redundant background information, increasing the computational load and hindering effective feature extraction from critical regions. Therefore, for each sample’s eight frames, the molten pool region was first segmented. Based on these segmentation results, the molten pool region was cropped from the original image with a 1:1 aspect ratio and then resized to 224 × 224 pixels without altering the pool’s original aspect ratio—this served as the input for deep learning models. This preprocessing scheme preserves image detail while improving processing speed by a factor of four without compromising accuracy. The resulting dataset, named WELDPOOL, was split into 60% training, 10% validation, and 30% test sets.

### 3.2. Test Environment

In our experiments, we used a desktop computer with a GeForce RTX 3060 GPU and a 12th-generation Intel(R) Core(TM) i9-12900H CPU running Windows 11. The WELDPOOL dataset was utilized, and the dataset was divided into the training, validation, and test sets in the ratio of 6:1:3. The experimental hyperparameters were set: the initial learning rate was 0.001; the weight decay was 0.0001; the momentum was 0.9; stochastic gradient descent was used to update the parameters; cosine annealing was used to adjust the learning rate; the number of training rounds was still set to 50; the batch size of each training iteration was set to 4, and the model was saved every three rounds with the best and final models. This ensures training stability while maximizing model convergence efficiency.

### 3.3. Performance Metrics

To objectively evaluate the effectiveness of the proposed model in identifying porosity defects, this study assesses the model from both performance and real-time capability perspectives. We use accuracy (Top-1 and Top-5), recall, and the F1-score as key metrics for performance evaluation. We also consider the number of parameters and computational complexity as evaluation criteria. Additionally, we include inference latency as a real-time evaluation metric.

### 3.4. Experiments

The experiments were conducted in the environment specified in Section 3.2. Firstly, we performed a training parameter analysis, testing the impact of data augmentation, fused image types, frame counts, and granularity levels on experimental performance to determine the final experimental parameters. Next, ablation experiments were carried out; specifically, experiments were conducted by separately excluding TSM, MAFA, and LDSSI auxiliary enhancement from the HIE-MSTARLNet model to illustrate the necessity and effectiveness of these technical modules. We then compared the specific performance of the model under different types of fused images and also compared it with several typical baseline welding defect detection models, demonstrating the superiority of the proposed approach. Additionally, confusion matrices of different methods were analyzed to further illustrate their effectiveness. We also tested normal molten pool samples under different conditions to analyze factors affecting performance. Finally, heatmap visualization tests were performed on molten pools from different welding state categories to analyze regions considered important by the model for defect identification.

## 4. Experimental Results

To speed up the training time, the MEHI as well as the frequency domain fusion images of the existing dataset are first computed here, and the PCA model that reduces the HOG and SIFT features to 100 dimensions is trained separately. In training, the multiple data loading form is used to directly input the original fusion pool image sequence and the generated MEHI or frequency domain fusion image, which eliminates the step of frequently calculating the MEHI and frequency domain fusion image; in the inference stage, the single data loading form is used to join the process of MEHI generation or frequency domain fusion image generation, which facilitates the calculation of the time spent on inference for each sample. And to further validate the interference resistance of the proposed model, data enhancement is carried out by adding random noise, flipping, rotating, and lighting changes to simulate as much as possible the various problems that may be encountered in a real welding scene.

### 4.1. Ablation Study

The core of the HIE-MSTARLNet model lies in the TSM, Multi-granularity Attention Feature Aggregation (MAFA), and long-term dynamic spatial structure information (LDSSI) aids. To this end, an ablation study was conducted to illustrate the necessity and effectiveness of these technical modules on the model. Specifically, the TSM, MAFA, and LDSSI-assisted augmentation from the HIE-MSTARLNet model, the long-term dynamic spatial structure information-augmented multi-granularity spatialtemporal representation learning network (LTDSS-MSTARLNet), the long-term dynamic spatial structure information-augmented single-granularity spatialtemporal representation learning network (HIE (HIE-SSTARLNet), the single-grained spatiotemporal representation learning network enhanced with long-term dynamic spatial structure information and time-shift information (SSTARLNet), and the time-shift information-enhanced multi-gradient spatiotemporal representation learning network (TS-MSTARLNet) were excluded. Then the performance of the three models is evaluated by accuracy and inference time, and the comparison results are shown in Figure 7.

From the figure, it can be seen that all three auxiliary enhancement tools, TSM, MAFA, and LDSSI, contribute to the performance of the model, and among them, TSM and MAFA contribute to a greater extent than the LDSSI auxiliary enhancement tool (i.e., TS-MSTARLNet has higher accuracy than the LTDSS-MSTARLNet model and the HIE-SSTARLNet model on the self-constructed dataset). The LTDSS-MSTARLNet model has higher accuracy than the HIE-SSTARLNet model on the self-constructed dataset, where MAFA contributes to a greater extent than the TSM (i.e., the LTDSS-MSTARLNet model has higher accuracy than the HIE-SSTARLNet model). However, both the MAFA module and the LDSSI-assisted enhancement tool significantly increase the inference time, whereas the TSM, being a plug-and-play module, adds little to no inference time. Among them, the MAFA module increases the inference time related to the number of its granularity levels, so it is important to minimize the granularity levels to reduce the inference time while satisfying high accuracy.

Overall, the ablation experiments of HIE-MSTARLNet validate the differentiated contributions of the three modules, TSM, MAFA, and LDSSI, to the model’s performance, providing a quantitative basis for the modular design of industrial vision models.

### 4.2. Comparative Result Analysis

To further illustrate the actual metrics performance of the HIE-MSTARLNet model using MEHIs and the HIE-MSTARLNet model using the frequency-domain fusion image (FDFI), the accuracy, precision, recall, F1 score, and inference time of the two models using different inputs are compared, as shown in Table 1,

Although inference times are comparable between the two methods, the HIE-MSTARLNet model based on the MEHI significantly outperforms the model using the FDFI in terms of accuracy and other key evaluation metrics. This advantage primarily stems from the superior feature representation provided by the MEHI approach. Specifically, the MEHI effectively integrates spatiotemporal dynamic information of the welding process by preserving historical edge details of the molten pool region across consecutive frames. This edge-history information not only captures subtle temporal variations in molten pool boundaries but also emphasizes minor structural differences in defect regions, facilitating more accurate defect identification. Additionally, the MEHI performs fusion directly in the spatial domain, thereby avoiding potential information loss and noise interference associated with frequency-domain transformations, enabling precise retention of essential local features at the original spatial resolutions. In contrast, the probable reason for the lower performance of the FDFI-based model is that additional noise may be introduced during the frequency-domain fusion process, which interferes with the model’s learning of effective features and weakens the high-resolution features of key local structures in the spatial domain due to the global frequency-domain transform, such as the formation of mosaic regions in the molten pool image, leading to a degradation of the model’s performance.

To demonstrate the superiority of the constructed weld defect detection model, the model was compared against several typical baselines of weld defect detection models, including a comparison of one class of models using static input data (i.e., image-based methods) and three classes of models using dynamic input data (i.e., RNN-based methods, 3-DCNN-based methods, and image fusion (IF) methods). For the three types of models using dynamic input data, the TSM, Multi-granularity Feature Aggregation (MGFA), and the attention module are sequentially inserted into the appropriate positions in the existing models to compare the performance of the model with the whole process of temporal feature extraction and the model with only temporal feature extraction at the end, as well as the performance of the model with the full process of temporal feature extraction and the performance of the model with only temporal feature extraction at the end. Specifically, all three types of models using dynamic data inputs use TSM; for the attention module, spatial and channel attention are used for the RNN-based and image fusion-based models, and spatial, channel, and temporal attentions are used for the 3-DCNN-based model. For a fair comparison, all models use MobileNetV2 as the backbone network and use consistent hyperparameter settings during the training phase. In particular, the optimal parameter values for model-specific hyperparameters (e.g., the time step and hidden cell number for the RNN-based approach and the input frame sequence length for the 3-DCNN-based and IF-based approaches) were filtered by a grid search and five-fold cross-validation, combining the typical values tested during previous training sessions and the actual test results in this experiment. Table 2 details the final comparison results of the four methods on the test dataset.

To avoid model chance, the values in the experiments are averaged over five cross-experiment runs as an evaluation metric. In terms of classification performance, these methods considering time-series information outperform image-based methods; feature-level fusion in time-series methods outperforms pixel-level fusion, and models fusing temporal information throughout the process also usually outperform methods fusing temporal information only at the end. Specifically, in these model comparisons, the proposed HIE-MSTARLNet model significantly outperforms other existing baseline methods by achieving 99.187%, 99.210%, 99.187%, and 99.190% in terms of accuracy (Acc), precision (Pre), recall (Rec), and F1 values, respectively. And, in terms of the inference time, methods that introduce temporal information generally take longer than single-image static methods (especially 3D-CNN-based methods), but HIE-MSTARLNet dramatically reduces the inference time while improving classification performance. And HIE-MSTARLNet has higher accuracy than MFVNet, but the inference time is a bit longer than MFVNet; the main reason comes from the generation time of MEHI, but in the actual deployment process, which is different from the MEHI inference experiments that generate the whole sample each time, the edge information of the first seven frames of the image can be cached at a specific time while actual receiving melting pool camera images, only the edge information of the current frame needs to be calculated, and weighted fusion needs to be performed; therefore, the inference time of the HIE-MSTARLNet model in practical applications can be greatly reduced.

To further analyze the model’s classification ability for different welding states, we select the optimal model from each of the five classes of methods (static image-based methods, RNN-based methods, 3D-CNN-based methods, pixel-level methods based on image fusion, and feature-level methods based on image fusion) and draw a normalized confusion matrix based on the classification results of the test set, as shown in Figure 8.

Each method has a small number of normal samples that are misclassified as pore defects, but the HIE-MSTARLNet model misclassifies fewer samples and is accurate in identifying pore defects, with only a small number of misclassified samples among the correct samples.

Therefore, to further analyze the sensitivity of the model in each sample, samples of normal molten pools with different conditions were tested. The results are shown in Table 3.

It can be observed that the HIE-MSTARLNet model has an accuracy of more than 99.7% in recognizing changes in the shape and size of the molten pool, and the accuracy in the splash interference scenario also reaches 98.374%. This indicates that the HIE-MSTARLNet model has improved anti-interference ability compared to the MFVNet model and can accurately recognize normal samples.

### 4.3. Visual Analysis

To further illustrate the interpretability of the proposed HIE-MSTARLNet model and to intuitively highlight its decision-making process in weld defect identification, we utilized class activation mapping (CAM) visualization. Specifically, molten pool images from four representative cases (CAM1–CAM4), covering both normal and porosity-defect scenarios, were selected for visualization, as shown in Figure 9. The first row corresponds to normal molten pools, and the second row corresponds to pools exhibiting porosity defects. The CAM results indicate that the model primarily focuses its attention on the central region of the molten pool surface, specifically the molten metal region below the welding wire. This central area exhibits notable variations in texture patterns and light spot distributions between normal and defective states, as opposed to solely relying on the outer contours of the molten pool, which has traditionally been the primary focus of previous studies. Notably, our visualizations also indicate that spatter around the molten pool is one potential interfering factor that might lead to misclassification. In practical industrial applications, these visual insights can effectively guide weld quality control by allowing operators and engineers to observe and understand critical discriminative features associated with defects directly. For example, noticing subtle differences in the highlighted central region may facilitate timely parameter adjustments or welding wire changes before significant defects occur. Thus, these CAM visualizations not only enhance the interpretability of the proposed model from a theoretical perspective but also offer actionable insights for industrial deployment, thereby bridging the gap between theoretical advances and real-world welding quality management.

### 4.4. Training and Parameter Analysis

The HIE-MSTARLNet model is first trained using MEHI images and raw sequence images, and the process is shown in Figure 10. It can be seen that the model trained based on the data after data augmentation exhibits more stable accuracy after convergence, although it converges more slowly than the model trained based on the original data. Moreover, the overall stability of the model improves significantly after data augmentation, and its performance stabilizes after about 35 epochs. To prevent overfitting, an early stopping strategy is used, where the number of training rounds is set to 10 rounds, and the model stops training at 42 rounds, so the model with 42 training rounds is used as the final model of the HIE-MSTARLNet model using MEHIs for the subsequent study.

Like the HIE-MSTARLNet model using MEHIs, in the training process of training the HIE-MSTARLNet model using frequency-domain fusion images, the same parameters, training strategy, and loss function are used, and the two models are trained using data-enhanced scenarios, which are shown in Figure 11. It can be seen that the HIE-MSTARLNet model using frequency-domain fused images converges more slowly and is somewhat less accurate than the HIE-MSTARLNet model using MEHIs. Therefore, the HIE-MSTARLNet model using MEHIs will be used as the final model in subsequent studies, and more detailed experiments will be conducted in the comparison experiments to illustrate the advantages of using the HIE-MSTARLNet model using MEHIs as the final model.

The effects of two key hyperparameters, the number of input frames and the granularity level, on the performance of the HIE-MSTARLNet model were investigated by varying the input frames and the granularity level, respectively, to test the classification accuracy of the model, as shown in Figure 12.

Consistent with the MFVNet model, the accuracy increases with the number of input frames. In addition, when the input frame sequence reaches a certain length, the accuracy increases with the increase in granularity level. Moreover, when the input reaches eight frames, accuracy stabilizes across different granularity levels, with no significant improvement from level 3 to level 4. To more accurately understand the influence of the number of sample frames and the number of granularity levels on the proposed model, the inference time of the proposed model is tested by changing the number of sample frames and the number of granularity levels, and the following experiments are set up to test the model. Firstly, the value of the input frames is fixed at eight, and the inference time of the model is tested at different granularity levels so that we can understand the effect of the change in granularity level on the inference time of the model, as shown in Figure 13a. It can be seen that when the granularity level increases, it has a significant effect on the reasoning time of the proposed model.

Then, the number of granularity level is fixed to three to test the inference time of the model when inputting different sample frames to understand the effect of different sample frames on the inference time of the model; in this experiment, the model structure is only tested for the time series with sample frames from 2 to 10. Figure 13b shows that when the input sample frames gradually increase, the model’s sample inference time also increases significantly.

In summary, while increasing both the number of sample frames and the number of granularity levels brings about an increase in the accuracy rate, it is accompanied by an increase in the inference time, which affects real-time detection in welding. Therefore, the number of input frames and the granularity level of the model were finally set to eight and three, respectively, to ensure accuracy while taking into account the inference time.

### 4.5. Stability Evaluation via Repeated Experiments and Cross-Validation

To assess the robustness of our model, we performed five independent training–validation–testing runs on the same data split, each with a different random seed. For each run, we recorded accuracy, precision, recall, the F1-score and the inference time. We then computed the sample’s mean (μ) and standard deviation (σ), as well as the 95% confidence interval (CI), under a normality assumption (The results are summarized in Table 4):(4)$${\mathrm{CI}}_{95\%}\;=\;\mu \pm 1.96\;\frac{\sigma }{\sqrt{5}}.$$

### 4.6. Multi-Defect Extension of the WELDPOOL Dataset

On top of the original WELDPOOL dataset, which contained only 1200 porosity defect samples, we added 678 burn-through and 678 lack-of-fusion samples (see Figure 14), resulting in a total of 2556 samples. We performed five independent training runs on this expanded dataset, achieving an average accuracy of 97.459%. Table 5 presents the mean (μ), standard deviation (σ), and 95% confidence intervals for accuracy, precision, recall, and the F1-score to evaluate the stability of the model’s performance.

## 5. Discussion

This study addresses challenges in welding defect detection, including the limited interpretability of deep learning models, restricted real-time performance, and interclass similarity issues, by proposing a hybrid-enhanced multi-granularity spatiotemporal representation learning algorithm (HIE-MSTARLNet). By integrating temporal dynamic modeling, handcrafted feature enhancement, and a multi-granularity attention mechanism, the model achieves enhanced interpretability and inference efficiency without sacrificing detection accuracy. Firstly, a Temporal Shift Module (TSM) is embedded within the MobileNetV2 backbone to progressively capture the short-term dynamic features of molten pool images, including pixel-level shifts at shallow layers and motion patterns at deep layers—effectively modeling the temporal evolution of the molten pool. Compared to traditional 3D convolutional or Transformer-based approaches, TSM significantly reduces computational complexity, meeting the real-time requirements of industrial scenarios. Simultaneously, we propose the Molten Pool Edge History Image (MEHI) generation method that combines the Histogram of Oriented Gradient (HOG) and Scale-Invariant Feature Transform (SIFT) features to extract long-term spatial structural information from the molten pool images. These features are compressed using Principal Component Analysis (PCA) for dimensionality reduction. By concatenating handcrafted and deep learning features, the model enhances interpretability regarding physical molten pool characteristics (e.g., texture anomalies and contour distortion) while mitigating the risk of overfitting caused by reliance on single-feature types. Next, we design a Multi-granularity Attention Feature Aggregation module (MAFA). This module first selects key interference-free frames via cross-frame attention, then generates features at different granularity levels through grouped pooling. Attention mechanisms are subsequently applied to these multi-granularity features, adaptively fusing spatiotemporal features (global structure and local details) and replacing traditional hybrid attention mechanisms. The experimental results demonstrate that MAFA significantly improves classification accuracy, reduces redundancy in multi-scale features, and optimizes computational efficiency. Parameter analysis determined the optimal number of input frames (eight frames) and granularity levels (three levels). Ablation experiments verified the effectiveness of the TSM, MAFA, and handcrafted feature modules. When evaluated solely on porosity defect detection, HIE-MSTARLNet achieved an accuracy of 99.187%. When burn-through and lack-of-fusion defects were added, HIE-MSTARLNet’s remained at 97.459%. Visual analysis further indicates that the model focuses on variations in light spots and texture within the central molten pool surface region, contrasting traditional contour-based attention, thus providing new insights for defect attribution.

While our model has shown high performance and good real-time capabilities on our custom dataset, some limitations remain. The current algorithm focuses on identifying potential porosity defects, and further exploration is needed to extend its applicability to identify a broader range of welding defects. Although we have extended beyond porosity defects to include burn-through and lack-of-fusion defects, the total dataset remains limited to 2556 samples. Deep learning models typically benefit from larger datasets to enhance their generalization and robustness. Future research should further explore model improvements, investigate a broader spectrum of welding defects and molten pool characteristics, and collect more diverse and larger-scale welding defect video datasets to improve model performance.

## Figures and Tables

**Figure 1 sensors-25-04656-f001:**
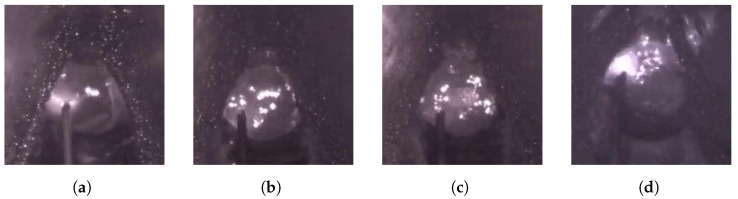
Two key challenges in welding molten pool image analysis: (**a**,**b**) show high interclass similarity between normal and porosity-defective molten pools; (**c**,**d**) illustrate significant intraclass variability among normal molten pools due to spatter, flow, and imaging conditions.

**Figure 2 sensors-25-04656-f002:**
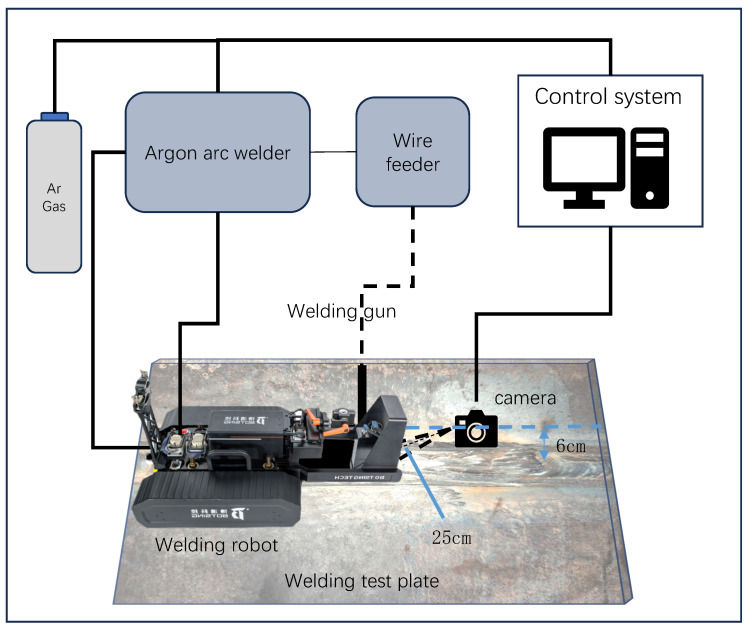
Schematic diagram of the molten pool vision system.

**Figure 3 sensors-25-04656-f003:**
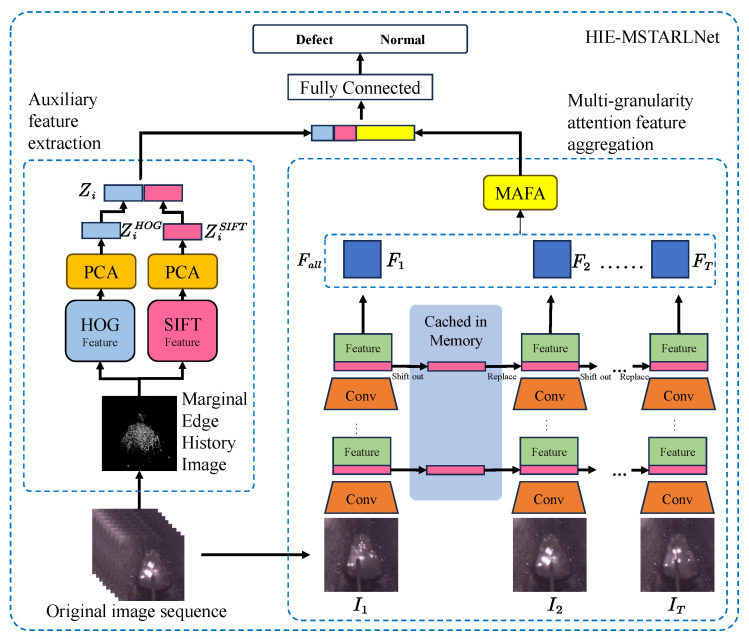
Hybrid information-enhanced multi-granularity spatiotemporal attentive representation learning network.

**Figure 4 sensors-25-04656-f004:**
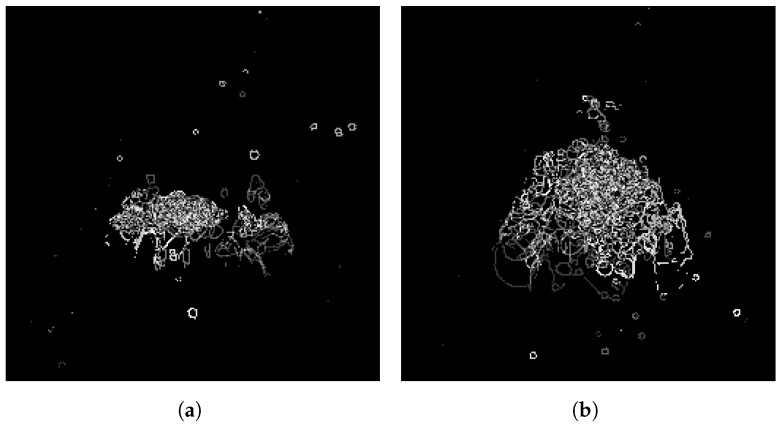
Edge history images of the molten pool at 8 time steps for different types of weld states. (**a**): normal molten pool; (**b**): porosity defects.

**Figure 5 sensors-25-04656-f005:**
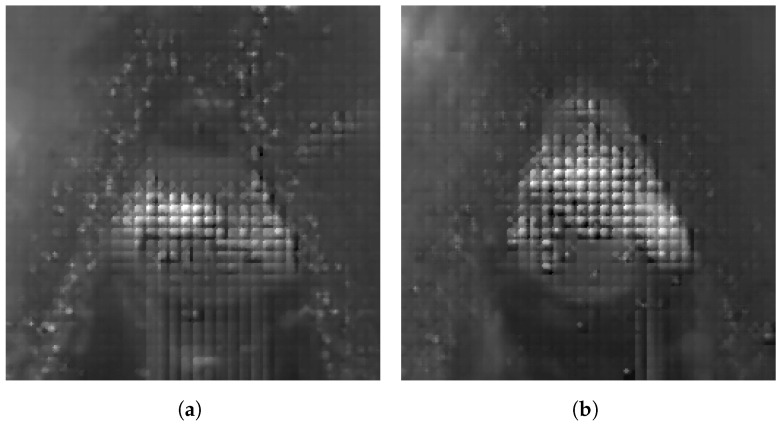
Frequency-domain fusion images of 8 time-step molten pools in different kinds of welding states. (**a**): normal molten pool; (**b**): porosity defects.

**Figure 6 sensors-25-04656-f006:**
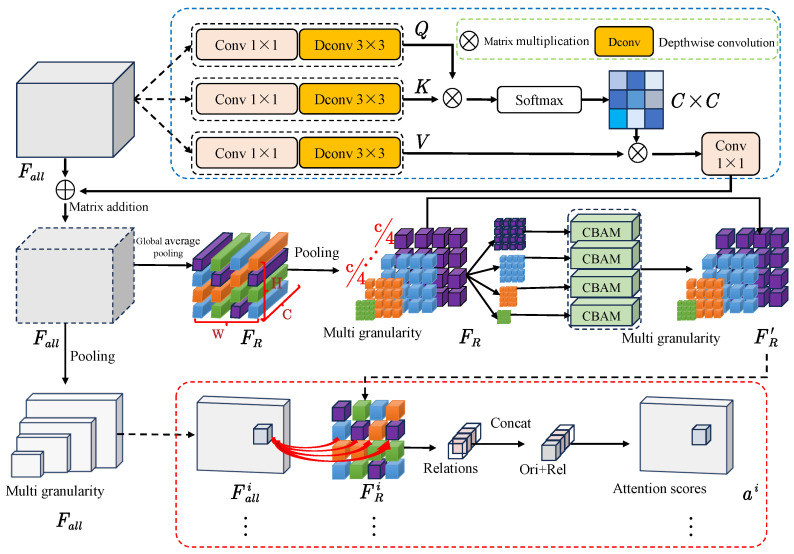
Multi-granularity Attention Feature Aggregation Module.

**Figure 7 sensors-25-04656-f007:**
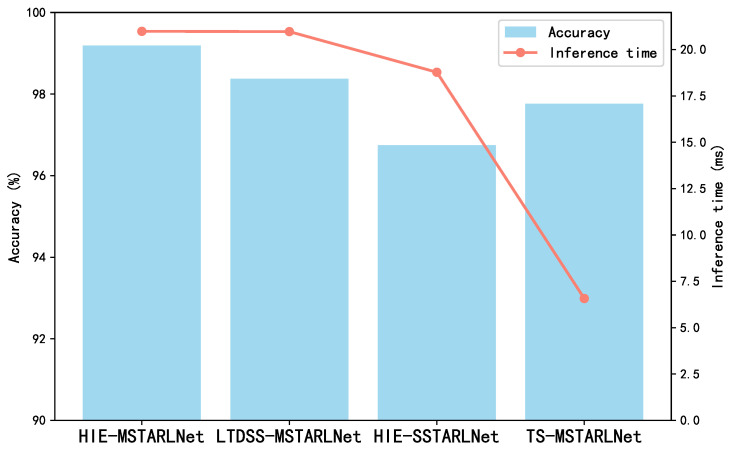
Ablation of the HIE-MSTARLNet model.

**Figure 8 sensors-25-04656-f008:**
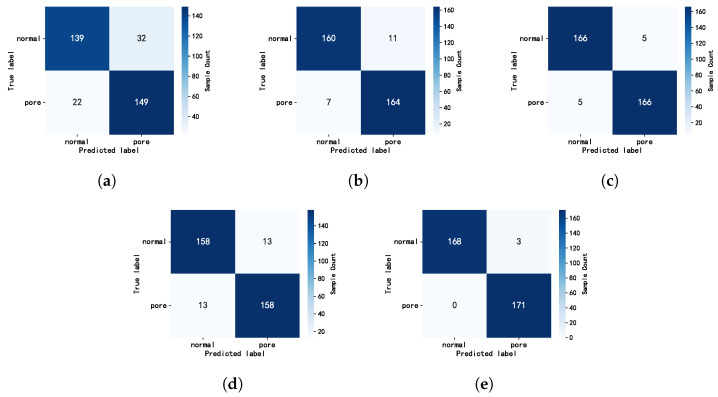
Normalized confusion matrices for different CNN models on the test dataset. (**a**) Image-based MobileNetV2; (**b**) MobileNetV2+LSTM (Attention+MGFA+TSM); (**c**) MobileNetV2-3D (Attention+MGFA+TSM); (**d**) MobileNetV2 (Attention+MGFA+TSM); (**e**) HIE-MSTARLNet.

**Figure 9 sensors-25-04656-f009:**
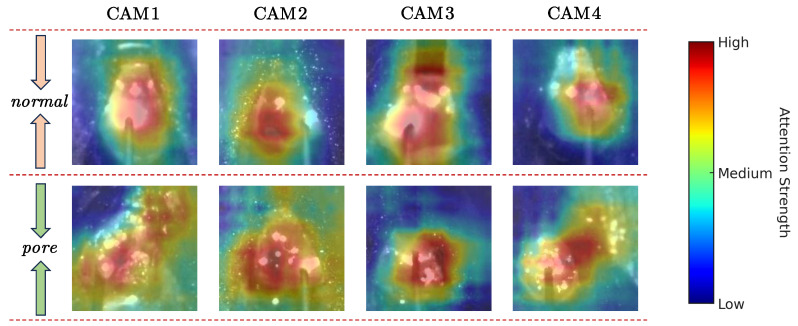
CAM of HIE-MSTARLNet in different welding state categories.

**Figure 10 sensors-25-04656-f010:**
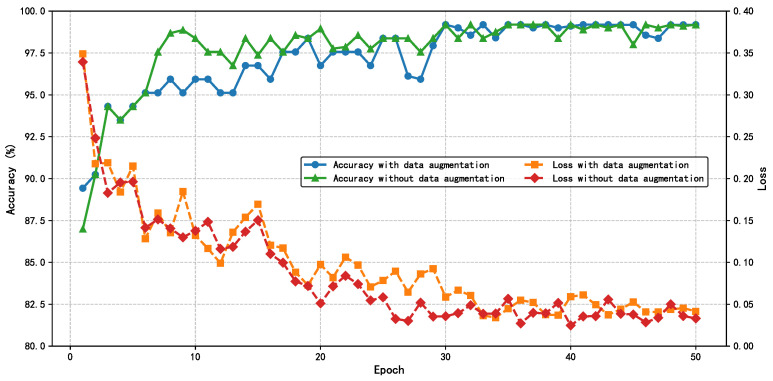
Training process of the HIE-MSTARLNet model.

**Figure 11 sensors-25-04656-f011:**
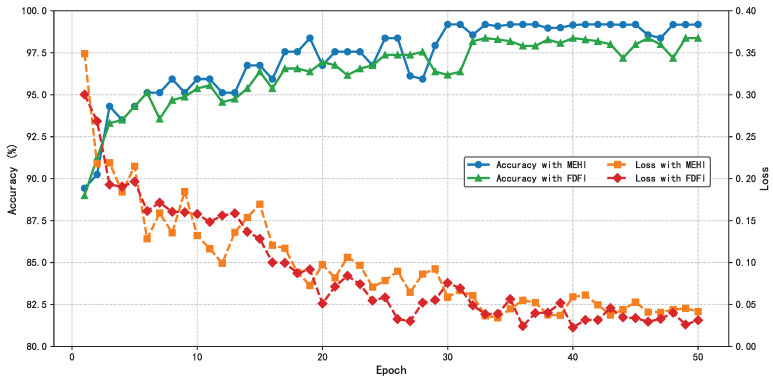
Comparison of training processes of the HIE-MSTARLNet model using MEHIs and frequency-domain fusion images.

**Figure 12 sensors-25-04656-f012:**
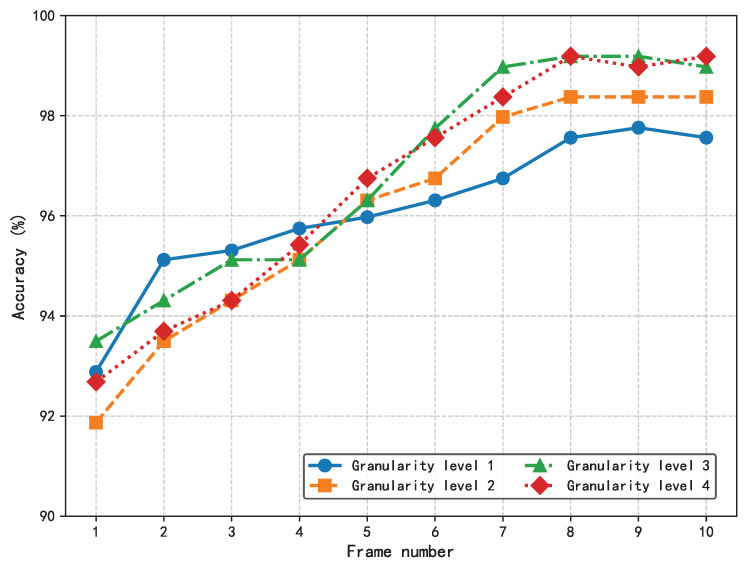
Effect of the number of sample frames and the number of granularity levels on the accuracy of the HIE-MSTARLNet model.

**Figure 13 sensors-25-04656-f013:**
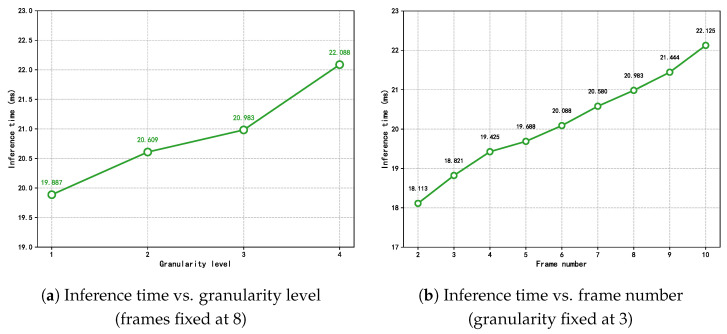
Effect of (**a**) granularity level and (**b**) the number of input frames on the inference time of the HIE-MSTARLNet model.

**Figure 14 sensors-25-04656-f014:**
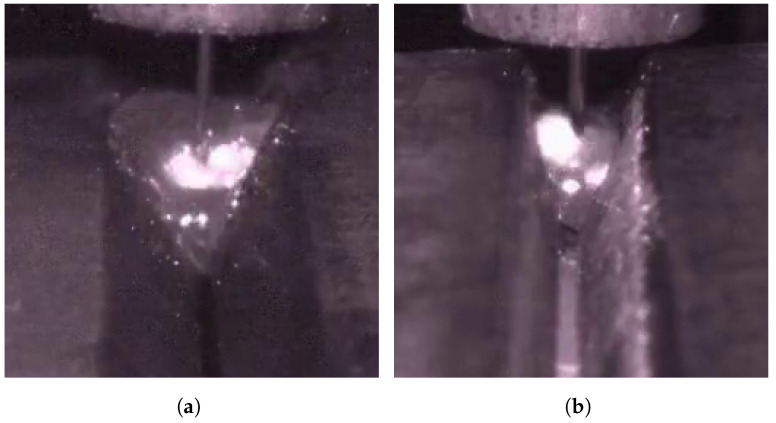
Sample images of two additional defect types: (**a**): burn-through; (**b**): lack-of-fusion.

**Table 1 sensors-25-04656-t001:** Comparison of results on datasets using MEHIs and the HIE-MSTARLNet model using FDFIs.

Data Type	Metrics				
	Acc (%)	Pre (%)	Rec (%)	F1 (%)	Time (ms)
FDFI	98.374	98.430	98.374	98.365	22.993
MEHI	**99.187**	**99.210**	**99.187**	**99.190**	**20.983**

Note: Bold values indicate the best performance across all metrics.

**Table 2 sensors-25-04656-t002:** Experimental results of different models on WELDPOOL datasets.

Model	Metrics				
	Acc (%)	Pre (%)	Rec (%)	F1 (%)	Time (ms)
image-based					
MobileNetV2	84.000	84.130	84.000	84.060	**2.1**
RNN-based					
MobileNetV2+LSTM	90.150	89.720	90.150	89.930	17.3
MobileNetV2+LSTM (TSM)	93.450	93.220	93.450	93.330	17.2
MobileNetV2+LSTM (MGFA)	94.450	94.590	94.450	94.520	19.0
MobileNetV2+LSTM (Attention)	93.850	92.860	93.850	93.350	18.7
MobileNetV2+LSTM (Attention+MGFA+TSM)	94.700	94.960	94.700	94.830	20.4
3DCNN-based					
MobileNetV2-3D	94.550	94.590	94.550	94.570	103.2
MobileNetV2-3D (TSM)	94.850	95.100	94.850	94.970	102.7
MobileNetV2-3D (MGFA)	96.450	96.780	96.450	96.610	104.8
MobileNetV2-3D (Attention)	94.750	94.630	94.750	94.690	133.8
MobileNetV2-3D (Attention+MGFA+TSM)	97.000	97.240	97.000	97.120	147.1
IF-based					
pixel-level					
MobileNetV2	86.500	85.230	86.500	85.850	3.3
MobileNetV2 (TSM)	89.000	89.040	89.000	89.020	4.2
MobileNetV2 (MGFA)	92.400	92.650	92.400	92.520	3.3
MobileNetV2 (Attention)	91.050	90.620	91.050	90.830	5.7
MobileNetV2 (Attention+MGFA+TSM)	92.400	92.580	92.400	92.490	8.1
feature-based					
HIE-MSTARLNet (Ours)	**99.187**	**99.210**	**99.187**	**99.190**	20.983

Note: Bold values indicate the best performance across all metrics.

**Table 3 sensors-25-04656-t003:** Results of HIE-MSTARLNet on normal molten pool samples with different conditions.

Categories	Conditions	
	#Case1	#Case2
splashes	99.954%	98.374%
s	99.927%	99.857%
size	99.892%	99.836%

**Table 4 sensors-25-04656-t004:** Five-run performance statistics (μ±σ) and 95% confidence intervals.

Metric	Accuracy (%)	Precision (%)	Recall (%)	F1-Score (%)	Time (ms)
μ	99.187	99.210	99.187	99.190	20.983
σ	0.042	0.037	0.045	0.031	0.321
95% CI	[99.129, 99.245]	[99.164, 99.256]	[99.121, 99.253]	[99.136, 99.244]	[20.688, 21.278]

**Table 5 sensors-25-04656-t005:** Five-run performance statistics (μ±σ) and 95% confidence intervals.

Metric	Accuracy (%)	Precision (%)	Recall (%)	F1-Score (%)
μ	97.459	97.514	97.447	97.478
σ	0.118	0.127	0.116	0.131
95% CI	[97.312, 97.606]	[97.356, 97.672]	[97.303, 97.591]	[97.315, 97.641]

## Data Availability

Data are contained within the article.

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
