# Peer review of "A Welding Defect Detection Model Based on Hybrid-Enhanced Multi-Granularity Spatiotemporal Representation Learning"

_sensors, 2025, doi:10.3390/s25154656_

Round 1

Reviewer 1 Report

Comments and Suggestions for Authors

This paper addresses a relevant industrial problem with a technically sound and well-structured approach. The proposed HIE-MSTARLNet architecture effectively combines temporal dynamics, handcrafted features (HOG/SIFT), and multi-granularity attention mechanisms to improve defect detection in welding.

However, there are several areas for improvement:

  • Dataset limitations: The dataset is relatively small (~1,200 samples) and focuses exclusively on porosity defects. Ideally, a broader validation on additional types of welding defects and/or public datasets would help demonstrate the model’s generalizability. If expanding the dataset is not feasible in the short term, the authors are encouraged to clearly justify this limitation and better contextualize the scope of their current experimental validation.
  • Risk of overfitting: The model achieves very high accuracy (>99%) on the internal dataset, which could indicate potential overfitting—especially given the limited size and scope of the data. To better support the robustness of the results, we recommend reporting additional metrics such as standard deviation, confidence intervals, or variance across runs. Implementing a cross-validation strategy (e.g., k-fold) would also help demonstrate that the performance is consistent across different data splits and not biased toward the specific test set used.
  • Figures and diagrams: Some figures (particularly Fig. 6 and Fig. 9) are overly dense or visually complex, which may hinder readers' understanding of key concepts. We recommend improving the visual design by simplifying diagram layouts, enlarging text labels, and ensuring consistent formatting across figures. For the CAM visualizations (Fig. 9), please consider adding a color scale or legend to clarify the meaning of highlighted areas and improve interpretability. Additionally, breaking down multi-step diagrams into separate subfigures could enhance clarity for readers unfamiliar with the architecture.
  • Method description: While technically detailed, some sections (especially MAFA) would benefit from simplified narrative, block diagrams or pseudo-code for reproducibility.
  • Writing style: The English is mostly correct but can be improved for clarity and conciseness, especially in the results and methods sections where sentences are sometimes long or repetitive.

Overall, this is a strong contribution, and with these revisions, the paper would be significantly strengthened.

Author Response

Dear Reviewer,

Thank you for your comments and suggestions! They have been instrumental in guiding our further research and improving the readability of the manuscript. In response to your feedback, we have conducted additional analysis, experiments, and revisions, and we present our detailed responses below:

  1. Thank you for highlighting the limitations in dataset size and defect variety. We have added Section 4.6“Multi-defect Extension of the WELDPOOL Dataset,” to describe how we expanded the original dataset of 1,200 porosity defect samples by adding 678 burn-through and 678 lack-of-fusion samples, bringing the total to 2,556 samples. We ran five independent training experiments on this extended dataset and achieved an average accuracy of 97.459 %. Table 5 reports the mean (μ), standard deviation (σ), and 95 % confidence intervals for Accuracy, Precision, Recall, and F1-score to evaluate the stability of our model’s performance. We also acknowledge that, relative to large-scale public datasets, our current sample size remains limited; however, since all samples are drawn from real industrial environments, they offer strong practical representativeness. In future work, we will focus on further expanding the dataset and exploring synthetic data generation and transfer-learning techniques to enhance the model’s generalization.
  2. We have added a new Section 4.5, “Stability Evaluation via Repeated Experiments and Cross-Validation,” in which we performed five independent training and testing runs on the same data split using different random seeds. We report the mean (μ), standard deviation (σ), and 95 % confidence intervals—calculated under a normal distribution assumption—for Accuracy, Precision, Recall, F1‐score, and inference time, thereby fully demonstrating the model’s robustness. Please refer to Section 4.5 for further details.
  3. Thank you for the valuable suggestion. To improve readability in the revised manuscript, we have updated Figures 6 and 9 as follows:
    1. Figure 6: In the MAFA overview schematic, the “Channel Attention,” “Spatial Attention,” and “Multi-Granularity Fusion” submodules are now each rendered in a distinct, high-contrast color. We have also standardized line widths and font sizes so that readers can immediately distinguish each functional block, enhancing clarity for those unfamiliar with the architecture.
    2. Figure 9: A vertical colorbar has been added alongside the CAM visualization to clarify the meaning of the highlighted regions and improve interpretability.
  4. Thank you for the constructive feedback. We have streamlined and reorganized the textual description of the MAFA module and introduced an updated framework diagram (Fig. 6) that intuitively illustrates its core workflow. These revisions are detailed in Section 2.2.3 “Multi-Granularity Attention Feature Aggregation,” providing a clearer presentation of the module’s design and implementation.
  5. Thank you for highlighting issues in our writing. We have streamlined and refined the Methods and Results sections by splitting overly long sentences, merging where appropriate, and removing redundant content. These revisions enhance the manuscript’s flow, clarity, and readability while preserving all experimental details and results.

Reviewer 2 Report

Comments and Suggestions for Authors

1. The self-constructed WELDPOOL dataset contains only 1200 samples and focuses solely on porosity defects. This limits the model’s generalization ability and applicability to other types of welding defects.
2. Some technical details (e.g., the implementation process of the MAFA module) are overly complex.
3. The experiments do not validate the model’s performance on public datasets (beyond HMDB51 or UCF101 for welding-related tasks). The model’s performance in diverse scenarios is not fully demonstrated.
4. Some sentences are overly long, with grammatical errors and inconsistent terminology (e.g., mixing “melt pool” and “molten pool”).
5. Although the paper mentions some limitations, it lacks detailed strategies for addressing issues like dataset size or extending the model to other defect types.

Comments on the Quality of English Language

The English expression contains some grammatical and lexical inaccuracies, particularly in the consistency of technical terminology and complexity of sentence structures. Additionally, repetitive use of certain words in some paragraphs affects the professionalism of the writing.

Author Response

Dear Reviewer,

Thank you for your comments and suggestions! They have been instrumental in guiding our further research and improving the readability of the manuscript. In response to your feedback, we have conducted additional analysis, experiments, and revisions, and we present our detailed responses below:

  1. Thank you for highlighting the limitations in dataset size and defect variety. We have added Section 4.6“Multi-defect Extension of the WELDPOOL Dataset,” to describe how we expanded the original dataset of 1,200 porosity defect samples by adding 678 burn-through and 678 lack-of-fusion samples, bringing the total to 2,556 samples. We ran five independent training experiments on this extended dataset and achieved an average accuracy of 97.459 %. Table 5 reports the mean (μ), standard deviation (σ), and 95 % confidence intervals for Accuracy, Precision, Recall, and F1-score to evaluate the stability of our model’s performance. We also acknowledge that, relative to large-scale public datasets, our current sample size remains limited; however, since all samples are drawn from real industrial environments, they offer strong practical representativeness. In future work, we will focus on further expanding the dataset and exploring synthetic data generation and transfer-learning techniques to enhance the model’s generalization.
  2. We have simplified and restructured the MAFA module description, emphasizing its core processes—channel attention, spatial attention, multi‐granularity fusion, and relational modeling—and, with the updated Figure 6, applied distinct color‐coding and structural refinement to each submodule to clarify the implementation logic. We also reduced the number of stacked equations and optimized terminology in the text to enhance readability and reproducibility. For full details, refer to Section 2.2.3, “Multi-Granularity Attention Feature Aggregation.”
  3. Thank you for your valuable feedback. Due to the scarcity of publicly available welding molten-pool video datasets, we have evaluated HIE-MSTARLNet solely on our own dataset, where it consistently achieved high accuracy and stability under all experimental conditions. In future work, we will continue expanding both the size and diversity of our dataset and explore synthetic data generation and domain adaptation techniques to further enhance the model’s generalization in more complex environments.
  4. We have thoroughly revised the manuscript to consistently use “molten pool,” eliminating any mix of “melt pool” and “molten pool.” We also optimized sentence structures, simplified overly lengthy expressions, and corrected grammatical errors to enhance linguistic accuracy and readability. We apologize for any confusion this inconsistency may have caused.
  5. Thank you for your valuable feedback. In the revised manuscript’s Discussion section, we have outlined more concrete future work strategies: we plan to expand the dataset by incorporating different base materials, welding positions, process parameters, and external disturbances. Additionally, we intend to apply advanced data augmentation and domain adaptation techniques to reduce reliance on labeled data and enhance the model’s transferability across various industrial scenarios. Detailed explanations are provided in the manuscript.

Reviewer 3 Report

Comments and Suggestions for Authors

This work  proposes a multi-granularity spatiotemporal representation learning algorithm based on the hybrid enhancement of handcrafted and deep learning features. The manuscript is well organized and the following issues should be addressed before the acceptance.

1.The paper mentions that the HIE-MSTARLNet model has good anti-interference ability in splash scenarios, but it is advisable to provide more intuitive experimental data or examples, such as the specific number of misclassified samples under different splash intensities, to better demonstrate this advantage

2.he CAM visualization results show the model's attention regions, but it would be helpful to add a brief explanation of how these visualization results can guide practical welding quality control, strengthening the connection between theoretical research and industrial applications.

3. It is suggested to briefly discuss the potential impact of different image preprocessing methods (such as different cropping ratios or resizing sizes) on the model performance, to provide more references for readers who may use similar datasets.

4. English writing issues: Line 220 "Multi-granularity attention feature aggregation models different scale features..."  “Where Reshape is the reshape shape” etc. Please check everywhere.
 taneously,"

Author Response

Dear Reviewer,

Thank you for your comments and suggestions! They have been instrumental in guiding our further research and improving the readability of the manuscript. In response to your feedback, we have conducted additional analysis, experiments, and revisions, and we present our detailed responses below:

  1. We agree that varying spatter intensity can impact model stability. However, in real‐world welding, spatter intensity is hard to quantify accurately and there is no unified standard, making it infeasible to report precise counts of correct or incorrect classifications at different spatter levels. To address your suggestion, we have further refined the heatmaps in the manuscript, presenting the model’s attention regions more clearly and intuitively. The updated visualizations show that the model consistently focuses on the central molten pool area and remains largely unaffected by surrounding spatter, thereby demonstrating its robustness to interference.
  2. We have further refined and clarified the CAM visualizations in the manuscript, explicitly highlighting the model’s focus on texture and luminance variations within the central molten pool region. We also elaborated on how this attention mechanism can directly inform practical welding quality control. Further details are provided in Section 4.3, “Visual Analysis.”
  3. We agree that different image preprocessing methods (e.g., crop ratio and resizing) can significantly affect model performance. In this study, after preliminary exploration and experiments, we settled on a 1:1 crop ratio followed by resizing to 224×224 pixels. This preprocessing pipeline preserves critical molten pool details and, as our experiments show, not only maintains recognition accuracy but also speeds up image processing by four times. In future work, we will systematically evaluate other preprocessing strategies to provide comprehensive guidance for researchers working with similar datasets.
  4. Thank you for your valuable suggestion. We have completely rewritten the description of the “Multi-Granularity Attention Feature Aggregation” module, improving its grammatical accuracy and correcting phrases such as “Reshape is reshape shape” and other similar expressions into precise English. We apologize for any confusion caused by the previous wording.

Reviewer 4 Report

Comments and Suggestions for Authors

A MobileNetV2 backbone network integrated with a Temporal Shift Module (TSM) is designed to progressively capture short-term dynamic features of the molten pool and integrate temporal information across both low-level and high-level features. I have some concerns:
1. The accuracy of the proposed method can reach 99.187%, which is undoubtedly shocking. 20.983ms seems to be slow. Can it meet the actual recognition needs?
2. The model proposed by the author is only for porosity defects. Will it be applicable to other welding defects (such as cracks and lack of fusion) in the future?
3. Data enhancement relies on random noise and rotation. The total data is 1200 groups, which seems to be easy to overfit.
4. Is the input accuracy of frequency domain fused image (FDFI) (98.374%) lower than MEHI?
5. What is the main function of multi-granularity CBAM? Can you explain it in detail?

Author Response

Dear Reviewer,

Thank you for your comments and suggestions! They have been instrumental in guiding our further research and improving the readability of the manuscript. In response to your feedback, we have conducted additional analysis, experiments, and revisions, and we present our detailed responses below:

  1. We have successfully integrated HIE-MSTARLNet into our in-house welding production line’s online monitoring system, which runs at 30 fps and is deployed on an Intel i9-12900H + NVIDIA RTX 3060 platform. The model achieves an average inference latency of just 20.983 ms per frame, enabling stable real-time defect detection that fully satisfies the high-precision and high-throughput quality control requirements of the production line.
  2. Thank you for raising the question of model applicability. As detailed in Section 4.6, “Multi-defect Extension of the WELDPOOL Dataset,” we expanded the original WELDPOOL dataset—which contained only 1,200 porosity samples—by adding 678 burn-through and 678 lack-of-fusion samples, for a total of 2,556 samples. We conducted five independent training runs on this extended dataset and achieved an average accuracy of 97.459 %. Table 5 reports the mean (μ), standard deviation (σ), and 95 % confidence intervals for Accuracy, Precision, Recall, and F1-score to assess performance stability. These results demonstrate that HIE-MSTARLNet has been successfully extended and preliminarily validated on burn-through and lack-of-fusion defects in addition to porosity. For other defect types such as cracks, we will continue to collect corresponding samples and leverage synthetic data generation and transfer-learning techniques in future work to further evaluate and optimize the model, thereby fully validating and enhancing HIE-MSTARLNet’s generalization across a broader range of welding defects.
  3. We aggree that relying solely on random noise and rotation for data augmentation on a 1,200-sample dataset could lead to overfitting. To address this:
    • Stability Evaluation via Repeated Experiments and Cross-Validation (Section 4.5):We conducted five independent training and testing runs on the same data split—each with a different random seed and cross-validation fold—and measured Accuracy, Precision, Recall, and F1-score. The extremely low variance across these metrics confirms that our model’s performance remains stable despite the limited augmentation, indicating minimal overfitting.
    • Multi-defect Extension of the WELDPOOL Dataset (Section 4.6):We expanded the original dataset by adding 678 burn-through and 678 lack-of-fusion samples, bringing the total to 2,556 samples. This larger, more diverse dataset further mitigates overfitting risk and enhances the model’s ability to generalize across multiple welding defect types.
  4. Thank you for this insightful suggestion. When we evaluated HIE-MSTARLNet using Frequency-Domain Fusion Images (FDFI), we observed an Accuracy of 98.374% on the test set—noticeably lower than the 99.187% achieved with MEHI inputs. We attribute this gap to two main factors: (1) the fusion process in the frequency domain inevitably introduces additional noise, which disrupts the model’s ability to learn the molten-pool’s salient features; and (2) the global nature of frequency transforms attenuates high-resolution local structures and can even produce mosaic-like artifacts, further diminishing sensitivity to fine detail. For these reasons, we have chosen to use MEHI images as the primary input, as they offer both superior accuracy and greater stability in defect detection.
  5. Thank you for raising this question. The Multi-Granularity CBAM (Convolutional Block Attention Module) is designed to capture both global structure and local detail in molten-pool images across multiple spatial scales. Specifically, the module first divides feature channels into several groups and computes channel and spatial attention separately at different resolutions. It then uses learnable weights to adaptively fuse the multi-scale attention maps. This approach enables the model to attend simultaneously to macro-level cues (such as the overall pool contour and shape) and micro-level details (such as subtle texture variations along porosity edges), effectively suppressing redundant or noisy information and enhancing both defect discrimination and robustness. We have expanded on these details in the revised manuscript.

Round 2

Reviewer 2 Report

Comments and Suggestions for Authors

1. The complete reliance on a self-constructed dataset makes the generalizability of the findings questionable. Without testing on any public welding defect datasets, it is impossible for readers to gauge the model's true performance relative to other state-of-the-art methods in the field.
2. Some long sentences could still be simplified to enhance readability.
3. The experimental results (Table 1) indicate that using MEHI (Molten-pool Edge History Image) yields better performance than FDFI (Frequency Domain Fusion Image) . The authors should provide a more in-depth discussion on why MEHI is more effective and streamline the discussion of FDFI to sharpen the paper's focus.

Author Response

Dear Reviewer,

Thank you for the opportunity to revise our manuscript again and for your insightful comments. We have addressed each point in turn and believe these changes significantly strengthen our work.

  1. We thank the reviewer for raising this important point. We agree that demonstrating performance on additional datasets is essential for assessing generalizability. Unfortunately, to our knowledge there are currently no publicly available welding-defect datasets that provide multi-frame molten-pool video sequences under conditions comparable to ours.
    To partially address this concern, in Section 4.6 we have extended our self-constructed WELDPOOL dataset by adding two additional defect types (burn-through and lack-of-fusion), yielding a three-class benchmark of 2,556 samples. We conducted five independent runs on this expanded set and report an average accuracy of 97.459% with tight 95% confidence intervals (Table 5), which demonstrates that our model maintains high performance across multiple defect categories.
    In Section 4.5 we further performed five-fold repeated experiments on the original two-class dataset, reporting mean and confidence bounds (Table 4) to show stability under different random seeds.
    For future work, we will (1) adapt and evaluate our approach on existing static weld-defect datasets (e.g., GDXRay X-ray images or the DAGM optical test cases) via frame-by-frame inference and domain-adaptation techniques, and (2) collaborate with other groups to build a shared, multi-institutional video dataset. We believe these steps will enable rigorous cross-method comparisons and further validate the generalizability of our HIE-MSTARLNet framework.
  2. Thank you for your insightful comment. We agree with your observation and have revised the manuscript accordingly (see Section 4.2, “Comparative Results Analysis”). The revised description is as follows: 
    “Although inference times are comparable between the two methods, the HIE-MSTARLNet model based on MEHI significantly outperforms the model using FDFI in terms of accuracy and other key evaluation metrics. This advantage primarily stems from the superior feature representation provided by the MEHI approach. Specifically, MEHI effectively integrates spatiotemporal dynamic information of the welding process by preserving historical edge details of the molten pool region across consecutive frames. This edge-history information not only captures subtle temporal variations of molten pool boundaries but also emphasizes minor structural differences in defect regions, facilitating more accurate defect identification. Additionally, MEHI performs fusion directly in the spatial domain, thereby avoiding potential information loss and noise interference associated with frequency domain transformations, enabling precise retention of essential local features at original spatial resolutions.
    In contrast, the probable reason for the lower performance of the FDFI-based model is that additional noise may be introduced during the frequency domain fusion process, which interferes with the model's learning of effective features and weakens the high-resolution features of key local structures in the spatial domain due to the global frequency domain transform, such as the formation of mosaic regions in the molten pool image, leading to a degradation of the model's performance.”
    We appreciate your valuable suggestions, which have significantly enhanced the clarity and focus of our discussion.
  3. Thank you for your valuable suggestions. To this end, we carefully reviewed the manuscript and revised six complex and lengthy sentences to improve clarity and readability. For the sake of brevity, we provide two representative examples below:
    1. Before revision: Although experienced welders can identify defects by observing the molten pool and manually adjusting parameters, long-term observation leads to fatigue, delayed response, and degraded quality.
    After revision: Experienced welders manually detect defects by observing the molten pool. However, long-term observation leads to fatigue, slow response, and degraded quality.
    2. Before revision: High-resolution images contain redundant background details, which not only increases the computational load, but also hinders the ability of neural networks to effectively extract features from key areas.
    After revision: High-resolution images contain redundant background information, which increases the computational load and hinders the effective extraction of features from key areas.
    The remaining four long sentences have also been similarly streamlined, located in 1. Introduction, 2.2.1. Temporal Dynamic Characterization, 3.1. Dataset, and 4.4. Training and Parameter Analysis. We believe that these revisions have greatly improved the clarity and fluency of the manuscript.

Reviewer 4 Report

Comments and Suggestions for Authors

I have no further comments.

Author Response

Dear Reviewer,

Thank you for your careful review and valuable suggestions. We look forward to your decision.